# Importance of non-stationary analysis for assessing extreme sea levels under sea level rise

Damiano Baldan[1], Elisa Coraci[1], Franco Crosato[1], Maurizio Ferla[1], Andrea Bonometto[1], Sara Morucci[1]

[1]Italian Institute for Environmental Protection and Research, ISPRA, Venice, Italy

Correspondence to: Sara Morucci (sara.morucci@isprambiente.it)

**Abstract.** Increased coastal flooding caused by extreme sea levels (ESLs) is one of the major hazards related to sea level rise. Estimates of return levels obtained under the framework provided by extreme events theory might be biased under climatic non-stationarity. Additional uncertainty is related to the choice of the model. In this work, we fit several extreme values models to two long-term sea level record from Venice (96 years) and Marseille (65 years): a Generalized Extreme Value distribution (GEV), a Generalized Pareto Distribution (GPD), a Point Process (PP), the Joint Probability Method (JPM), and the Revised Joint Probability Method (RJPM) under different detrending strategies. We model non-stationarity with a linear dependence of the model's parameters from the mean sea level. Our results show that non-stationary GEV and PP models fit the data better than stationary models. The non-stationary PP model is also able to reproduce the rate of extremes occurrence fairly well. Estimates of the return levels for non-stationary and detrended models are consistently more conservative than estimates from stationary, non-detrended models. Different models were selected as being more conservative or having lower uncertainties for the two datasets. Even though the best model is case-specific, we show that non-stationary extremes analyses can provide more robust estimates of return levels to be used in coastal protection planning.

## 1. Introduction

Coastal zones are extremely vulnerable to extreme sea levels (ESLs; Kron, 2013). Exposure to coastal flooding damage is projected to increase in the future (Jongman et al., 2012) due to higher frequency, magnitude, and duration of extreme sea levels (Tebaldi et al., 2021; Devlin et al., 2021). The mean sea level rise is among the causes of this increase (Menéndez and Woodworth, 2010; Marcos et al., 2009). The design of structures to protect coasts from flooding (minimizing e.g. damages to infrastructures and coastal erosion) relies on the knowledge of ESLs that are likely to occur with a given probability (Boettle et al., 2016). Extreme events theory provides a theoretical background to fit historical extremes with specific probability distribution functions (Coles et al., 2001), and is widely used for estimating the probability of occurrence of ESLs. However, non-stationarity poses some challenges to the development of solid estimates of such return levels.

The results of extreme value theory are valid under the assumptions of independence and stationarity of extremes (Khaliq et al., 2006). Here, stationarity means that all the realizations of the extremes in the data record are generated from the same distribution (Coles et al., 2001). While independence is satisfied with a proper selection of extremes from the dataset, stationarity is often assumed but not verified (Khaliq et al., 2006). However, several sources of non-stationarity can affect sea level data: changes in coastal morphology, low frequency climatic variability, and climate change (Salas and Obeysekera, 2014). The estimation of return levels from stationary models might not be appropriate (e.g. less conservative) because of the implicit assumptions that the characteristics of the extremes remains the same in the future (Caruso and Marani, 2022; Razmi et al., 2017; Dixon and Tawn, 1999; Salas and Obeysekera, 2014; Haigh et al., 2010; Ragno et al., 2019). Two approaches are commonly used to cope with non-stationarity. Detrending the sea level data with annual or long term mean sea levels is a common practice to remove long term signals in the mean of the dataset (Bernier et al., 2007; Tebaldi et al., 2012; Mentaschi et al., 2016). Alternatively, the parameters of the probability distribution function that generates the extremes can be explicitly modelled as dependent from some covariates (Méndez et al., 2007; Grinsted et al., 2013; Cid et al., 2016; Sweet and Park, 2014; Razmi et al., 2017). However, clear indications on which approach suits better non-stationary conditions are still missing.

The choice of the proper method to conduct the extreme sea level analysis is also a challenge. Several methods exist, both direct, based on fitting theoretical Probability Distribution Functions (PDFs) to the data, and non-direct, relying on mixtures of empirical and parametric PDFs. Extreme values in the data can be defined as either maxima over uniform blocks of data, or values that exceed a defined threshold (Coles et al. 2001). Several theoretical PDFs were derived accordingly: the Generalized Extreme Value distributions (Mudersbach and Jensen, 2010), the Generalized Pareto distributions (Wahl et al., 2017), and the Point Process (Boettle et al., 2016). Indirect methods such as the Joint Probability Method, or the Revised Joint Probability Method (Pugh and Vassie, 1978) also exist. These methods decompose the sea level in the tide and surge components. Different methods might be more or less suited (in terms of explained and residual variance, see section 2.3.6) to accommodate non stationary data, and might lead to different estimates of extreme sea level probabilities (Wahl et al., 2017; Razmi et al., 2017). However, a comparison of the suitability of different direct and indirect methods for modelling non-stationarity is currently missing.

Using two long-term sea level time series from Venice (96 years, NE Italy), and Marseille (65 years, Southern France) with different extents of non-stationarity, this this paper aims at: (i) compiling information on the existing direct and indirect methods for extreme sea levels estimation; (ii) assessing which parametric method and detrending approach best accommodates non-stationary conditions; and (iii) comparing return level and return period estimates from different parametric and non-parametric methods. We perform all the analyses using three different detrending approaches.

## 2. Methods

### 2.1 Tide gauges location

The city of Venice and its lagoon are exposed to the risk of flooding due to extreme sea levels (Ferrarin et al., 2022). The tide regime is semi-diurnal, with mean tidal range from 50 cm during neap tide to 100 cm during spring tide (Umgiesser et al., 2021): such values are among the highest ranges measured in the Mediterranean sea. Compared to other sites in the northern Adriatic sea, the Venice Lagoon experienced higher sea level rise due to the combined effects of subsidence and eustatism (+ 2.5 mm year$^{-1}$ in the last 150 years, Biasio et al., 2020; Zanchettin et al., 2021). The current long-term mean sea

level is about 30 cm above the local 1897 reference (named Zero Mareografico di Punta della Salute, ZMPS: average sea level for the period 1885-1909 measured at the Punta della Salute gauging station). As a result, an increase in the frequency and magnitude of ESLs causing flooding of the city of Venice was recorded (Umgiesser et al., 2021). The events with the highest recorded sea levels occurred on November 4th, 1966 (+ 194 cm), and November 12th, 2019 (+ 189 cm, Lionello et al., 2021).

On the contrary, the area where the Marseille tide gauge is located has a lower tidal range (around 10 cm, Fig S1), and is located on a stable geological background, with a relative sea level rise of + 1.1 mm y$^{-1}$ in the last 150 years (Letetrel et al., 2010; Wöppelmann et al., 2014). Marseille data are referred to the nautical chart datum (NCD, *Zero Hydrographique*), the sea level corresponding to the lowest tide, and is 32.9 cm below the national datum (IGN1969, average sea level for the period 1885-1897 measured at Marseille gauging station, Wöppelmann et al., 2014). The long term mean sea level was 35

cm above the NCD in 1903 and 50 cm above the NCD in 2017.

### 2.2 Tide gauge data

We used sea level data recorded by the tide gauge station located in Venice (gauge name: *Punta della Salute*) covering the period 1924 – 2019. Data from 2020 onwards are affected by the activation of a storm surge barrier system that prevents ESLs to propagate inside the Venice lagoon (MOSE) and therefore were not included in the analysis. The float-operated tide

gauge is located inside a still well; measurements were recorded mechanically until 1988 and electronically from 1989 onwards. Until 1989, semidiurnal maxima and minima are available (4 measurements per day); then data were recorded hourly in the period 1989 - 1994, every half hour in 1995 - 2006, every ten minutes in 2007 - 2019. We resampled all data recorded after 1989 to an hourly resolution with Pugh filters. We used a filter with 27 coefficients for 10-minutes data, a filter with 18 coefficients for 15-minutes data, and a filter with 12 coefficients for 30-minutes data (Pugh, 1987). The data

have no gaps; a total record length of 96 years was used to fit the models. To calculate long-term mean sea level before 1924, we used yearly mean sea level data from other tide gauge stations active in the city of Venice (and thus affected by the same subsidence rate as *Punta della Salute*) whose records cover the period 1885 – 1922 (namely: *Campo Santo Stefano*, *Arsenale*, and *Punta della Salute – Canal Grande*; for details see Zanchettin et al., 2021).

Hourly sea level data recorded at Marseille are available for the time period 1849 – 2017. Measurements were performed with a float-operated tide gauge until 1988, with an acoustic sensor for 1989 – 2008, and with a radar sensor from 2009 onwards. The measurements were recorded mechanically until 1988 and electronically from 1989 onwards (Wöppelmann et al., 2014). A total record length of 65 years (spanning 1903 - 2017) was used to fit the models (incomplete years were discarded).

### 2.2.1 Data detrending approach

We used two different approaches for detrending the sea level data before fitting the models: a) we removed from each sea level observation the yearly average mean sea level (hence after: MSL detrending); b) we removed from each sea level observation the sea level average calculated over the previous 19 years (hence after: MSL_L detrending), to remove long term fluctuations due to interferences between lunar precession and solar activity (Valle-Levinson et al., 2021); c) we used non detrended data to fit the models (hence after: NDT).

### 2.3 Extreme Values distributions

Extreme events are defined as events with a low probability of occurrence (Coles et al., 2001). Given a set of independent and identically distributed random variables $X_1, \ldots, X_n$, with parent distribution $F$, a probability distribution function describing the occurrence probability of extreme values can be derived with two approaches. The Block Maxima (BM) approach considers the distribution of the maxima of the set $X_1, \ldots, X_n$ over blocks of length $n$: $M_n = max\{X_1, \ldots, X_n\}$ and assesses $Pr(M_n < z)$, i.e. the probability that the random variable $M_n$ is greater than $z$. The use of sufficiently large blocks ensures that the maxima are independent (Méndez et al., 2007). The Peaks Over Threshold (POT) approach assesses $Pr(X > u + y \,|\, X > u)$, i.e. the probability that the random variable $X$ exceeds a sufficiently high threshold $u$ by the value $y$. When fitting POT, an appropriate threshold needs to be selected to properly model excesses as extremes (Zhang et al., 2000). In this work, we used a block length of one year to extract BM to fit the GEV models. We selected the threshold for POT models (GPD and PP) with a two-step approach. First, data above the 99th percentile were selected, and events separated by more than a fixed time window were considered independent and retained. We used a time window of 78 hours for both Venice and Marseille, consistent with the value used in other studies in the Mediterranean (Marcos et al., 2009). This value also corresponds to the average decay time of seiches, the lowest-frequency sub seasonal oscillation in the Northern Adriatic sea (Masina and Ciavola, 2011; Raicich et al., 1999). Second, we fitted multiple POT models based on different thresholds, and we selected the lowest value that ensures the stability of the GPD and PP parameters (see section 2.3.2). This procedure ensures that the threshold excesses can be properly modeled as extremes, and eq. 2 holds (Coles et al., 2001). For the Venice data, thresholds of 100 and 80 cm are appropriate to select POT for non-detrended and detrended data, respectively, yielding 319 POT for NDT, 284 for MSL, and 359 for MSL_L. For the Marseille data, thresholds of 85 and 40 cm are appropriate to select POT for non-detrended and detrended data, respectively, yielding 203 POT for NDT, 207 for MSL, and 223 for MSL_L.

### 2.3.1 Generalized Extreme Value Distribution

The BM distribution depends on F, the parent distribution of the random variables in each block via: $G(z) = Pr(M_n < z) = F^n(z)$, converging to the generalized extreme values (GEV) distribution when n is large enough (Coles et al., 2001):

$$G(z) = exp\left[-\left\{1 + \xi\left(\frac{z-\mu}{\sigma}\right)\right\}_+^{-1/\xi}\right] \tag{1}$$

where $a_+ = max(a, 0)$, μ is the location parameter (proportional to the first-order moment of the distribution), σ is the scale parameter (always positive, proportional to the second-order moment of the distribution), and ξ is the shape parameter that determines the type of distribution function: the heavy-tailed Frechet ($\xi > 0$), the upper-bounded Weibull ($\xi < 0$), and the limit-case Gumbel ($\xi \to 0$).

### 2.3.2 Generalized Pareto Distribution

The POT distribution depends on F, the parent distribution of the random variables via: $H(y) = Pr(X > u + y \mid X > u) = (1 - F(u+y)/(1 - F(u))$, with $y=z-u$, converging to the Generalized Pareto Distribution (GPD) when the threshold is large enough (Coles et al., 2001):

$$H(z) = 1 - \left[1 + \xi\left(\frac{z-u}{\sigma_u}\right)\right]_+^{-1/\xi} \tag{2}$$

where $u$ is the threshold, $\sigma_u$ is the GPD scale parameter dependent on the threshold, and $\xi$ the shape parameter that
determines the type of the distribution function: heavy-tailed Pareto ($\xi > 0$), upper bounded Beta ($\xi < 0$), with the Exponential as limit-case ($\xi \to 0$). When BM are GEV-distributed, POT is theoretically expected to follow a GPD with the same shape parameter and scale depending on the GEV parameters $\sigma_u = \sigma + \xi(u - \mu)$ (Gilleland and Katz, 2016). This property can drive the selection of an appropriate threshold u. First, multiple GPD distributions are fitted to different sets of data obtained varying the threshold. Then, the parameters are plotted as a function of the threshold. For sufficiently high
thresholds, the theoretical approximation yields and the parameters are independent of the threshold value. The minimum threshold that meets this requirement is then selected (Coles et al., 2001).

### 2.3.3 Point process approach

The occurrence of POT can be modeled also as a point process. Under stationary conditions, the process follows a Poisson distribution (Coles et al., 2001; Menéndez and Woodworth, 2010):

$$O(k) = Pr(X = k) = \frac{\lambda^k e^{-k}}{k!} \tag{3}$$

where λ is the rate of the process (number of events over a reference time period). The process rate depends on the GEV parameters (Gilleland and Katz, 2016; Boettle et al., 2016; Cid et al., 2016):

$$\lambda = \left[1 + \xi\left(\frac{z-\mu}{\sigma}\right)\right]_+^{-1/\xi} \tag{4}$$

When location and scale are not constant (e.g. a dependence from a covariate is introduced), the process rate is not constant over time and the point process is non homogeneous (Cebrián et al., 2015). The probability of occurrence of extremes in a non-homogeneous point process is not constant over time, hence this model is appropriate for modelling extremes whose occurrence frequency is not constant in time (Coles et al., 2001).

### 2.3.4 Joint Probability and Revised Joint Probability methods

Unlike the methods mentioned above, the joint probability method (JPM) is non-parametric. The JPM is based on the decomposition of the sea level z in the tide (x) and surge (y) components (Pugh and Vassie, 1978). The probability distribution of the sea level $P(z)$ results from the convolution of the distributions of the tide and the surge:

$$P(z) = \int_{-\infty}^{+\infty} P_T(z - y) \, P_S(y) \, dy \qquad (5)$$

where $= x + y$, $P_T(x)$ is the distribution of the tide, and $P_S(y)$ is the distribution of the surge (both obtained from hourly records). The tide and the surge interaction is significant only in shallow waters (Prandle and Wolf, 1978), and can be considered independent in most practical applications (Pugh and Vassie, 1978).

Two limitations of the JPM are that consecutive sea levels are assumed to be independent, and that the upper tail of the empirical surge distribution is biased by the lack of observations of extremes. As a result, the JPM cannot produce ESLs estimates for sea levels higher than the combination of the highest tide and surge (Batstone et al., 2013; Tawn et al., 1989). The revised Joint Probability Method (RJPM) aims at improving both issues. First, an extremal index that accounts for dependencies in the sea level data is introduced ($\theta^{-1}$, units: hours). The extremal index is used as a correction factor in the return period calculation based on $P(z)$ (see section 2.3.7), and is defined as the average number of measurements an extreme sea level cluster is usually composed of (Tawn et al., 1989). Second, the RJPM fits the surge distribution with an extreme values distribution to smooth the empirical distribution, for projections beyond the highest measured surge (Tawn et al., 1989). Both GEV and GPD have been used to this end (Baranes et al., 2020; Batstone et al., 2013; Enríquez et al., 2022).

The tidal component of the mean sea level used in the JPM was calculated with the 'oce' package (Kelley, 2018) in the R computing environment v4.1.2 (R Core Team, 2021), using the yearly detrended sea level data (MSL), 7 harmonic constants (M2, S2, N2, K2, K1, O1, P1) for Venice, and 24 harmonic constants for Marseille (MSM, MM, MSF, MF, Q1, O1, NO1, PI1, P1, S1, K1, J1, 2N2, MU2, N2, NU2, M2, L2, T2, S2, K2, MN4, M4, MS4; Wöppelmann et al., 2014). The surge was calculated as the difference between the sea level observation and the corresponding tide. We used 1990-2019 hourly data from for Venice and 1968-2016 for Marseille (record length of 30 years for both stations) The same time series were used to calculate the tidal coefficients for tide estimation.

For the JPM, we used all the tide and surge data from the sea level decomposition to generate the empirical frequency distribution over classes with a width 10 cm. The maximum theoretical sea level (sum of maximum tide and maximum surge) falls within the highest class. Then, we calculated the discrete convolution between the two histograms (see table 1 in Pugh and Vassie, 1980). For the RJPM, we fitted a Gumbel distribution function to the annual maxima of the surge

(following Tawn et al., 1989). Then we defined surge classes of width 10 cm and calculated the probability of the surge extremes to fall in each class as the integral of the fitted Gumbel distribution calculated over each class. After the convolved distribution was calculated, we used the probability of sea level falling in each sea level class to calculate the return periods (Pugh and Vassie, 1978; Marcos et al., 2009). We then corrected the estimated return levels with the extremal indices. We found extremal indices of 5.5 hours and 13 hours to be appropriate for Venice and Marseille, respectively.

### 2.3.5 Models fitting

We used the package 'ExtRemes' (Gilleland and Katz, 2016) to fit the parametric models (GEV, GPD, PP) based on the Maximum Likelihood criterion (Castillo et al., 2005; Coles et al., 2001).

### 2.3.5 Stationarity and parameters dependence

Both BM and POT approaches require the modelled random variables to follow the same parent distribution F. Non stationary conditions can be modelled by including covariates in the GEV, GPD, and PP parameters (Méndez et al., 2007). For instance, a linear dependence of location ($\mu$) and scale ($\sigma$) parameters can be assumed from the covariate c and can be expressed as (Coles et al., 2001):

$$\mu(c) = \mu_0 + \mu_1 c \tag{6}$$

$$log(\sigma(c)) = \sigma_0 + \sigma_1 c \tag{7}$$

where the logarithm on the scale parameter in eqn. 7 is used to constrain the scale parameter to positive values.

### 2.3.6 Comparing different models configurations

The likelihood ratio test is employed to assess whether the inclusion of a covariate in the model formulation improved significantly the fit. Two nested competing models $M_0 \subset M_1$ can be compared using the deviance statistic (Coles et al., 2001). For example, $M_1$ can be a model that whose parameters depend on covariates, while $M_0$ a model whose parameters do not depend from covariates. The deviance is expressed as:

$$D = 2\{l_1(M_1) - l_0(M_0)\} \tag{8}$$

where $l_1(M_1)$ and $l_0(M_0)$ are the maximized log-likelihoods of models $M_1$ and $M_0$, respectively. The model $M_0$ has by definition a lower complexity than $M_1$, which is the case when covariates on the model's parameters are added to $M_1$. High deviance values support the hypothesis of $M_1$ explaining a larger variation in the data than $M_0$ (likelihood ratio test). The hypothesis is rejected when $D > c_\alpha$ where $c_\alpha$ is the $(1 - \alpha)$ quantile of a $\chi_k^2$ distribution, where $k$ is the difference in dimensionality between $M_1$ and $M_0$ (i.e. the difference in the number of parameters).

### 2.3.7 Return levels estimation

The return period is defined as: $Tr(z) = [1 - G(z)]^{-1}$, where G is the Probability Distribution Function for the GEV, GPD, or PP models (Caruso and Marani, 2022), or the empirical sea level probability from the JPM and RJPM (Tawn et al., 1989). In practice, the extreme levels of the random variable are calculated as a function of the return period via the PDF quantiles (Coles et al., 2001). In a non-stationary analysis, the model's PDF is not constant in time (Fig. 1), and the quantiles are not uniquely determined. To allow for the comparison of estimated return levels from non-stationary models, in this work we first fixed the covariates values, and then calculated the quantiles of the resulting probability distribution function. For the JPM and the RJPM we included the extremal index as a correction factor in the estimation of the return period $Tr(z) = \theta^{-1}[1 - G(z)]^{-1}$.

### 2.4 Data analysis

Before fitting the models, we employed a Mann-Kendall test to check if BM and POT resulting from different detrending strategies follow a temporal trend. Additionally, we used linear models and quantile regressions (75th quantile) to relate BM and POT with the mean sea level, and used the significance of the regressions as indication for stationarity.

To check if the inclusion of non-stationary covariates can improve the models (objective ii), we fitted different configurations of GEV, GPD, and PP models to the full dataset (96 years). We fitted: a) models without covariates; b) models with the location linearly depending on the yearly mean sea level; and c) models with location and logarithm of the scale linearly depending on the yearly mean sea level. We used the likelihood ratio test (eq. 8) to assess whether the inclusion of mean sea level-dependent parameters improved the fit significantly.

To check visually the dependence of parameters from the mean sea level, we fitted stationary GEV, GPD, and PP models (i.e. without covariates on the scale and location parameters) to BM and POT subsets using a 30-years moving time window. We can assume that data sampled in a 30-years window can be considered stationary. We tested for the presence of a trend in the fitted parameters with a Mann-Kendall test. We plotted the sequence of stationary parameters together with non-stationary ones as a mean to visually check the uncertainty related to parameters estimation.

The PP models were further validated by comparing the process rate (eq. 4) and the empirical rate of POT exceedances (number of excesses per year) with a Pearson's correlation test.

After fitting the models, we compared the estimates of the return level for different return periods (objective iii). For the non-stationary models, we first calculated the location and scale parameters with a yearly mean sea level of + 35 cm in Venice and + 52.4 in Marseille (equal to the 2000 - 2019 long-term mean sea level for the two stations). Once the model's parameters were fixed, we calculated the sea levels corresponding to return periods of 2, 20, 100, and 200 years. Estimates of return levels from models fitted to detrended data were added back the long term mean sea level. This additive procedure is simplified and neglects the non-linear interactions between future mean sea level and the occurrence of extremes (Arns et al., 2015, 2017).

Finally, we derived the curves from non detrended, non-stationary models under different covariates values. For Venice, we used + 0 cm; + 25 cm (annual mean sea level in 1966, the year of the largest ESL on record); + 35 cm (annual mean sea level for 2019, the last year used in the analysis); and + 51 cm (expected annual mean sea level in 2050 under IPPC scenario SSP2-4.5, Garner et al., 2021; Masson-Delmotte et al., 2021). For Marseille, we used + 54 cm (annual mean sea level for 2019), and + 71 (expected annual mean sea level in 2050 under IPPC scenario SSP2-4.5).

## 3. Results

Regarding the data used to fit the models, the Mann-Kendall tests detected a significant trend for the non-detrended BM in Venice, a marginally significant trend for the detrended BM, and no trend for POT (Fig. 2). No trends were recorded for BM or POT in Marseille (Fig. S2). We found evidence for a dependence of the median BM on the mean sea level for both detrended and non-detrended data in Venice, while little support for a trend was recorded in Marseille. In Venice, the median

POT, and the upper POT quantile were significantly dependent from the mean sea level only for the MSL_L detrending method (Table 1).

After fitting the models, the likelihood ratio test for Venice data shows that the inclusion of the covariate (mean sea level) improves the fit significantly for the location ($\mu$) parameter of both GEV and PP for NDT and MSL_L data, and only for GEV for MSL data (Table 2). The addition of a dependence on the scale ($\sigma$) parameter was marginally significant for the

GPD for NDT and MSL_L data. The inclusion of the dependence from the scale on the PP improved the fit only for MSL_L data (Table 2). In Marseille, the inclusion of the covariate improved the fit for the location for GEV and PP for NDT, and for the location of PP for MSL_L data.

Models validation for Venice showed that the location parameter dependent on the covariate well reproduces the temporal trends of the corresponding stationary parameters obtained from the time-window analysis in GEV and PP. Smaller

improvements are observed for Marseille, where most of stationary models well reproduce the parameters trends (Figure S3). Additionally, the PP models estimated the occurrence rate of threshold exceedances in Venice in good agreement with those calculated from the POT data (Table 3).

The return levels estimated by non-stationary models for Venice were in the range 133 – 146 cm for a return period of 2 years, 169 – 184 cm for 20 years, 192 – 203 cm for 100 years, and 198 – 218 cm for 200 years (Fig. 4, Table S1). Estimates

of 100-years return levels for non detrended models with covariates were in the range 169 – 181 cm, while for detrended models without covariates were higher (186 – 187 cm). Models that include covariates on the location showed an increased extreme estimate for smaller return periods (< 10 y for GEV, and < 3 y for PP, Fig. 4, S4), with higher discrepancies for non detrended data. The return levels estimated by non-stationary models for Marseille were in the range 102-114 cm for a return period of 2 years, 128-141 cm for 20 years, 139-153 cm for 100 years, and 144-158 cm for 200 years (Table S2). Estimates

of 100-years return levels for non detrended models with covariates were in the range 143-147 cm, while for detrended models without covariates were higher (156-153 cm).

Finally, we compared how the return levels for return periods of 2, 20, 100, and 200 years differ among models (Fig. 5, Table S1). Among stationary models, the GPD yields conservative estimates for 2 years and the GEV is more conservative for 20 and 100 years for all detrending configurations. Among models with covariates on the location, GEV yields higher return levels estimates. Among non-stationary models fitted to non-detrended data, GPD models with covariate on the scale yield conservative estimates for all return periods. Estimates from GEV models with covariates on location and scale fitted to detrended data are more conservative for 20, 100, and 200 years. The JPM and RJPM yields projections that are in agreement with parametric models. Return levels from models without covariates fitted to non detrended data were consistently the less conservative for all return periods and both Venice and Marseille. The highest differences between detrended, non detrended and stationary models were higher for short return periods. Among all the analysed methods, in Venice the GEV with covariate on the location, the JPM, and RJPM yield the most conservative estimates of return levels for longer return time (>50 yr), while for return time of 2 and 20 years the RJPM is less conservative than other methods. A similar behaviour is observed in Marsille for RJPM. Differently, in Marsille the JMP provides less conservative return level for all return times. A consistent behaviour was observed when stationary models fitted to data covering 30 years were compared with JPM and RJPM (Figure S5).

The direct methods show varying uncertainty in the prediction of return levels (Figure 5). In both Venice and Marseille, the GEV with covariates on the location has the highest uncertainty (12 cm in Venice and 15 cm in Marseille) for the 2 years return period, and the PP without covariates the lowest uncertainty (7 cm both in Venice and Marseille). In Venice, the PP with covariates on the location has the lower uncertainty for return levels of 20, 100, and 200 years (15, 20, and 25 cm, respectively). Non detrended models without covariates and detrended models have similar uncertainty (slightly lower for GEV). In Marseille, the GEV fitted to non detrended data has the lowest uncertainty for return levels of 20, 100, and 200 years (13, 23, and 27 cm, respectively).

Extrapolations of non detrended, non-stationary models for the future showed that estimates of future ESLs are strongly influenced by the future mean sea level (Fig. 6). Events that currently have a return level above 200 y will have return levels < 30 y (for GEV and GPD) and < 50 y (PP) already in 2050 for Venice. For Marseille, events that currently have a return level above 200 y will have return levels < 30 y (for GEV and GPD) and < 100 y (PP) already in 2050.

## 4. Discussion

### 4.1 Including non-stationarity in extreme events modeling

Our results show that most of the fitted ESL models benefit from the inclusion of covariates on either the location and the scale parameters when using non detrended data. Highest improvements in the fit occurred for the Venice data, that have an higher non-stationariety than Marseille. We used only the yearly averaged mean sea level as covariate to build simple models, but other predictors can be used, depending on the objective of the study. For instance, the North Atlantic Oscillation Index, the Arctic Oscillation, the East Atlantic/Western Russia Oscillation index can be used to include a

dependence from climate (Menéndez and Woodworth, 2010). Where climatic predictors are missing, seasonality effects can be included e.g. with an harmonic dependences from the yearly Julian day (Méndez et al., 2006). Other predictors could include global and regional meteorological parameters, which could influence storm surges intensities and frequencies (Grinsted et al., 2013). A dependence from time can be also included (Mudersbach and Jensen, 2010). However, particular care should be used in the choice of the predictors. Complex models can be useful for explaining historical pattern, but might be of little utility for future projections. For instance, bias could arise due to uncertainties in predictor's future trajectories, or to future predictor's values out of the ranges used to calibrate the models. In this regard, simpler models can be helpful for future projections when clear links between extremes occurrence and specific predictor's classes are established (Schuwirth et al., 2019).

In this work, we used the mean sea level as covariate because of the strong link with storm surges occurrences (Lionello et al., 2021). Our results show that mean sea level-dependent location of both GEV and PP models improve the ESLs fit for both Venice and Marseille. The location parameter is the first-order moment of the extremes distributions. The inclusion of a linear dependence from the mean sea level translates rigidly the distribution function towards higher (positive slope) or lower (negative slope) values without affecting the shape of the distribution. GEV and PP models also marginally improved with a dependence on the scale. The scale parameter relates to the second-order moment of the distribution (the "spreading"). A dependence of the scale parameter from the mean sea level could suggest an influence on the variability in the magnitude of storm surges. In shallow area an higher sea level corresponds to lower dissipation of the tidal energy, yielding higher ESLs (Arns et al., 2017). In the Venice lagoon, this factor might be influenced also by the morphological transformations that the Venice Lagoon underwent during the 20th century and that might have affected the dynamics of the tide propagation (Caruso and Marani, 2022). Different explanations for this pattern are possible. For instance, the North Atlantic Oscillation Index (NAO), not included in this analysis, might act as a latent variable: negative NAO phases in the Mediterranean basin can lead to increases in monthly mean sea levels and in the number of storms (Cid et al., 2016).

Overall, this work shows how including non-stationarity in extreme events analysis can support an improved understanding of extreme events. Including dependences from the mean sea levels allows for flexible forecasts of ESLs also under sea level rise scenarios.

## 4.2 Comparison of the models

The significant covariate dependencies could be also influenced by the type of data. The BM data in Venice show clear increasing trends, which were captured by the GEV model. BM could be extracted with different methods, such as monthly blocks, or for r-largest yearly values. A global analysis (Wahl et al., 2017) showed that the annual maxima is the more conservative method (i.e. yields higher return period estimates). However, this aspect should be checked as part of a sensitivity analysis from case to case. POT data do not have a trend in the mean or in the higher quantiles neither in Venice or Marseille, thus should yield models that are less affected by non-stationarity. However, a trend in the frequency of occurrences of POT (Ferrarin et al., 2022) was observed in Venice, which might invalidate the homogeneity assumptions of

GDP and PP models. The non-homogeneity of the POT distribution can be mitigated by introducing a dependence of the threshold from a covariate (Roth et al., 2012). However, using a non-constant threshold introduces a significant uncertainty that might result in biased estimates of the return levels (Agilan et al., 2021). On the contrary, the PP explicitly models the

rate of threshold exceedances: the detected significant dependence of location from the mean sea level implies a process with non-constant occurrence rate (i.e. a non-homogeneous process, eq. 6, Cid et al., 2016). The ability of the PP models to predict the changes in ESLs occurrence frequency with sea level rise is of particular relevance in Venice, where a system of movable gates was recently built to disconnect the lagoon from the sea and prevent the flooding of the city during ESLs (Umgiesser, 2020).

While all the parametric methods improved with the inclusion of non-stationarity, the JPM and RJPM are the methods that should be least influenced by non-stationarity, since the methodology requires to detrend the data before the calculation of tide and surge histograms. However, as the residual trend on detrended BM for Venice shows, the removal of the mean sea level might not be sufficient to make the series stationary. Thus, also estimates of the return level with the JPM might be biased. Estimations of return levels for long return periods are not possible due to the lack of surge and tide events that are

needed to populate the extremal classes of the distribution. In our analysis, JPM allows for estimating return periods corresponding to levels of + 233 cm in Venice (corresponding to the sum of the maximum recorded tide, + 57 cm, the maximum recorded surge, + 141 cm, and the current mean sea level, + 35 cm) and + 163 cm in Marseille (tide: + 20 cm, surge: + 89 cm, current mean sea level: + 54 cm), but for shortest series, this limitation might be stronger. Another limitation of the JPM and RJPM as implemented in this paper is the lack of information on uncertainty. The development of

multivariate ESLs distributions could solve this issue (Ferrarin et al., 2022), but applications that include non stationarity are still limited.

Some parametric models were improved by the inclusion of covariates on the location, with a stronger influence on models fitted to non detrended data. Particular care should be taken when detrending data prior to the model fit, as this action implicitly assumes that the mean sea level is the main responsible of data non-stationarity, and higher order interactions are

neglected. In shallow area this could not be the case (Arns et al., 2017). Thus, inclusion of covariates on the model's parameters could perform better than detrending in such cases. With our data, the use of the mean sea level as covariate results in a rigid translation towards higher return levels for GEV and PP plots due to the significance of the location dependence. The effect on GPD is also in a change in slope due to the significance of the scale parameter. However, data from different gauging stations might show different behaviours. For instance, sites where the sea level variability increases

with mean sea level might show a significant dependence in the scale parameter also in GEV and PP.

Our results show that the models that have lower uncertainty and the models that yield the most conservative estimates of return levels are different between Venice and Marseille. Even though we highlighted that the difference between the two datasets is the extent of non-stationarity, other factors can affect the selection of a good model: the relative importance of tide and surge (Dixon and Tawn, 1999), the tidal regime, the location of the tide gauge, the record length, and the presence

of outliers (Haigh et al., 2010). A similar analysis on a dataset covering a wider range of sites would allow to consistently link the best performing methods to the characteristics of the sea level data.

## Conclusions

In this paper, we fitted different extreme value models to long-term sea level data for Venice and Marseille. We show that including non-stationarity in the analysis of extreme events improves the fit of most of the models. Among direct methods,

for return periods longer than 20 years, the Point Process with a dependence of the location from the mean sea level is the most conservative in Venice. The Generalized Extreme Values distribution with a dependence of the location from the mean sea level is the most conservative in Marseille. Among indirect methods, the Revised Joint Probability Method yields results that are comparable with the most conservative methods for return periods longer than 100 years for both Venice and Marseille. Among direct methods, the Generalized Extreme Values Distribution fitted to detrended data has the lowest

uncertainty for return levels estimation in Venice. The Point Process with a location dependence has the lowest uncertainty for return levels estimation in Marseille for return periods longer than 20 years. Overall, we show that non-stationary extremes analyses can provide more robust estimates of return levels to be used in coastal protection planning.

## Aknowledgements

This work was supported by the EU-INTERREG project ADRIACLIM (grant number IT-HR 10252001)

**Data availability**

Sea level data from Venice used in this work are freely available in the web portal of the Italian Institute for Environmental Protection and Reaserch (ISPRA): https://www.venezia.isprambiente.it/rete-meteo-mareografica. Sea level data from Marseille were accessed through the IOC portal at: http://www.ioc-sealevelmonitoring.org/ssc/stationdetails.php?id=SSC-mars.

**Author contributions**

SM and DB designed the study and implemented the models. FC and EC collected the data. DB led the writing of the manuscript with inputs from all co-authors.

**Competing interests**

The authors declare that they have no conflict of interest.

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

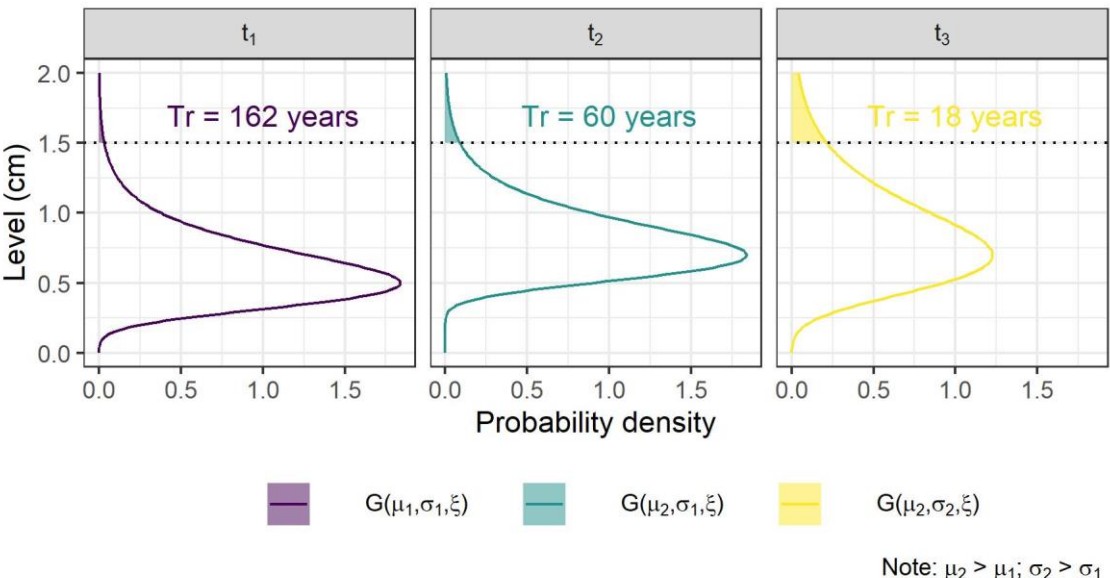

**Figure 1: Example of the effects of curves parameters on the return period estimation. GEV curves with different location ($\mu$) and scale ($\sigma$) parameters corresponding to three time periods are represented. The shape ($\xi$) parameter is kept constant The return period is calculated based on the highlighted area (see section 2.3.7). Different location and scale yield different return period estimates. Under non-stationary conditions, the curve's parameters change with time.**

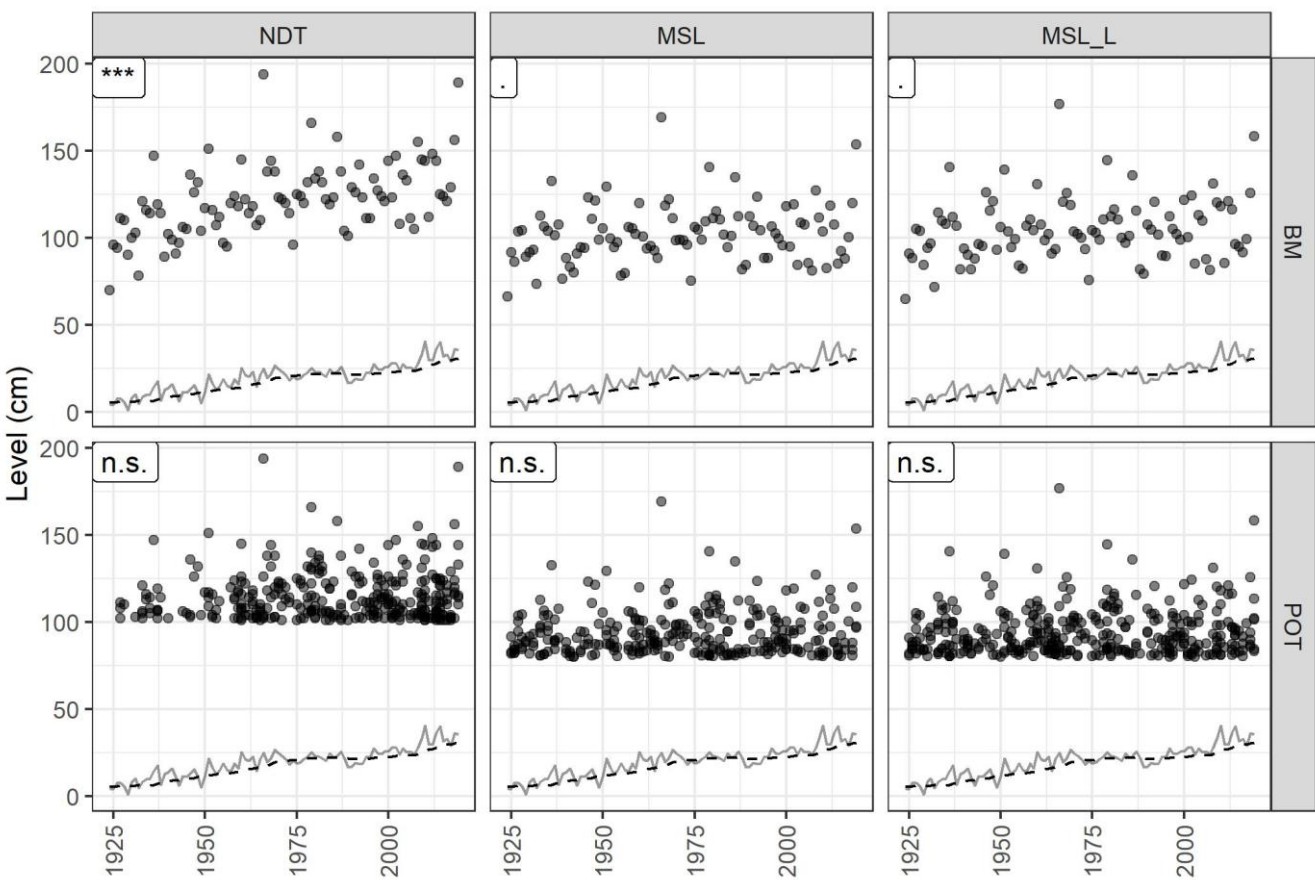

**Figure 2: Venice data used to fit the models. See Fig. S2 for Marseille data. Plots are grouped vertically according to the detrending method (MSL: mean sea level, MSL_L: long term mean sea level, NDT: non detrended), and horizontally according to the maxima typology (BM: block maxima, POT: peak over threshold). The text in the label on the top-left corner of each plot shows the significance level of the Mann-Kendall trend test (n.s.: non significant; .: $p < 0.1$; *: $p < 0.05$; **: $p < 0.01$; ***: $p < 0.001$). The continuous line represents the mean sea level value; the dashed line represents the long-term mean sea level.**


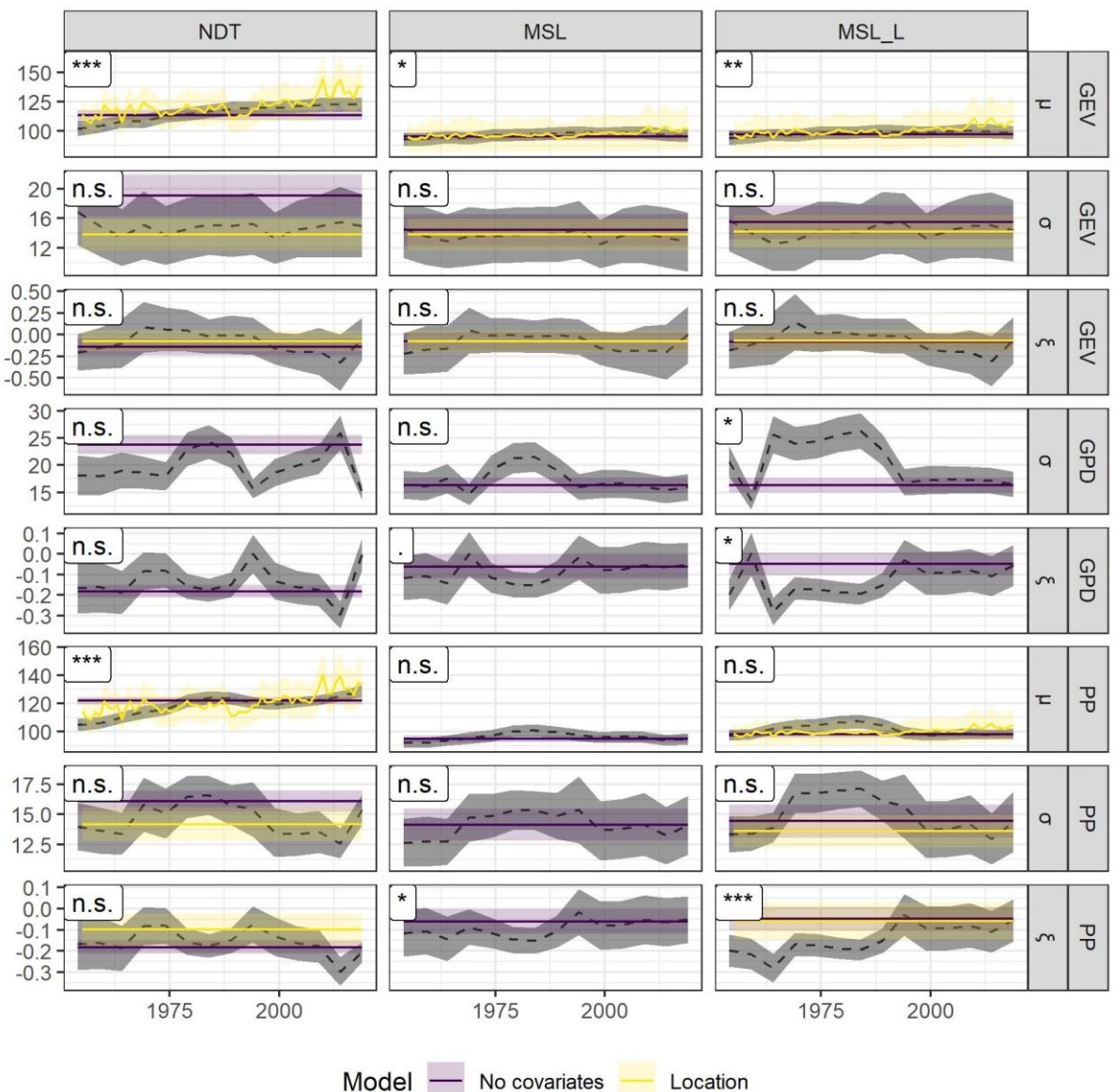

**Figure 3: Comparison between the parameters estimated in the time window analysis (dashed line; the grey envelope represents the uncertainty of the parameters from the time window analysis) and the parameters estimated by different models configurations over the full data length. Here only results for Venice are reported; see figure S3 for Marseille. Parameters from all the configurations of GEV, GPD and PP that do not include covariates are showed. Parameters from models with covariates are showed only if models improve significantly the fit (see Table 2 for the likelihood test). The shape $\xi$ is included in the figure, but no covariates dependence was tested for this parameter. The horizontal axis represents the final year of the time window. Plots are grouped vertically according to the detrending method (MSL: mean sea level, MSL_L: long term mean sea level, NDT: non detrended), and horizontally according to the distribution function (GEV: generalized extreme values, GPD: generalized pareto, PP: point process). The text in the label on the top-left corner of each plot shows the significance level of the Mann-Kendall trend test on the parameters from the time-window analysis (n.s.: non significant; .: $p < 0.1$; *: $p < 0.05$; **: $p < 0.01$; ***: $p < 0.001$).**

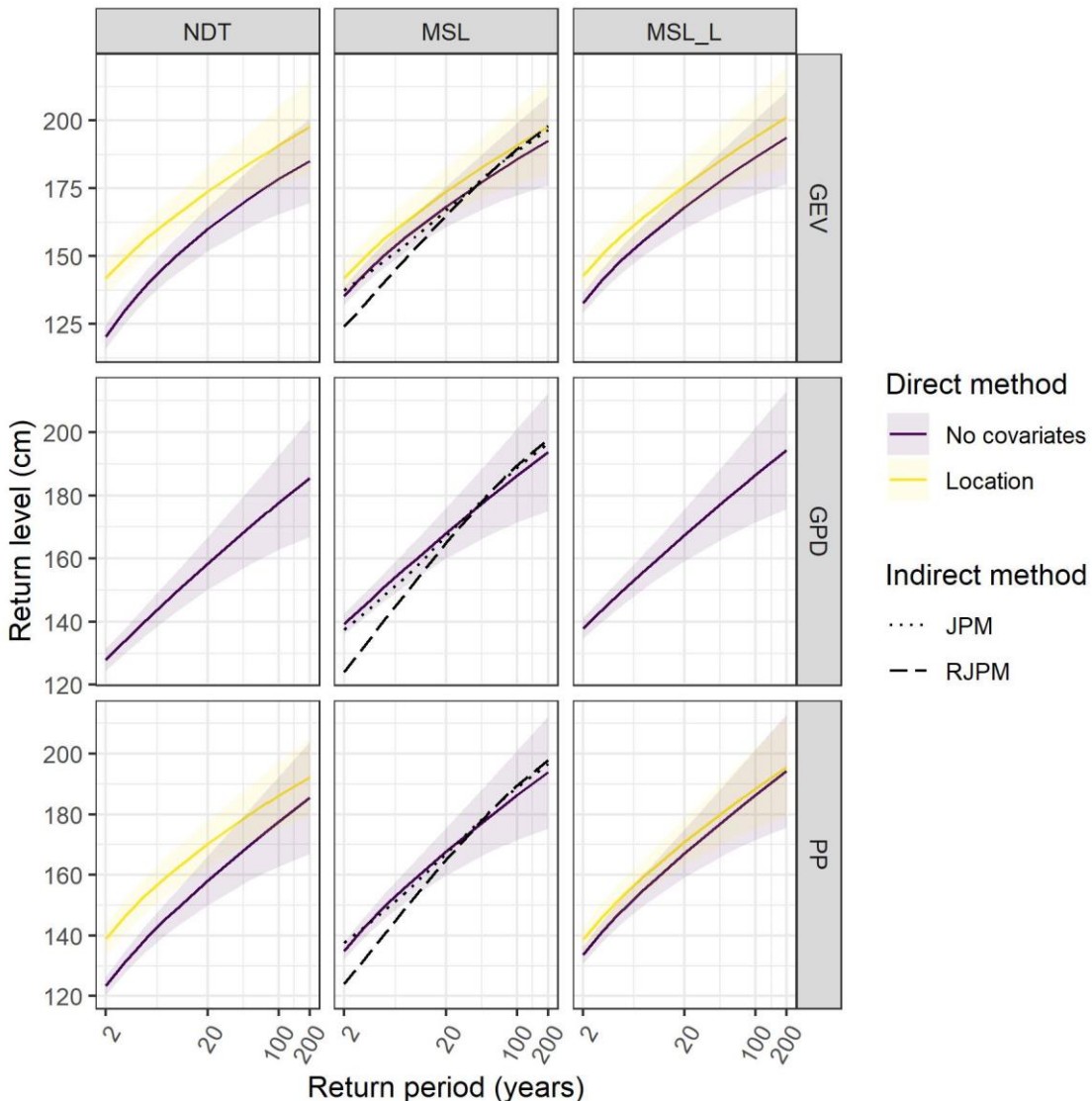

**Figure 4: Return level plot actualized to 2019 for Venice. See figure S4 for Marseille. Plots are grouped vertically according to the detrending method (MSL: mean sea level, MSL_L: long term mean sea level, NDT: non detrended), and horizontally according to the distribution function (GEV: generalized extreme values, GPD: generalized pareto, PP: point process). The dashed line is the empirical return level for the joint probability method (JPM). Curves are color-coded based on the model configuration. Note: horizontal axis is logarithmic. Return level curves for direct models with covariates are reported only if the addition of the covariate improves the fit significantly (p < 0.01; see Table 2).**


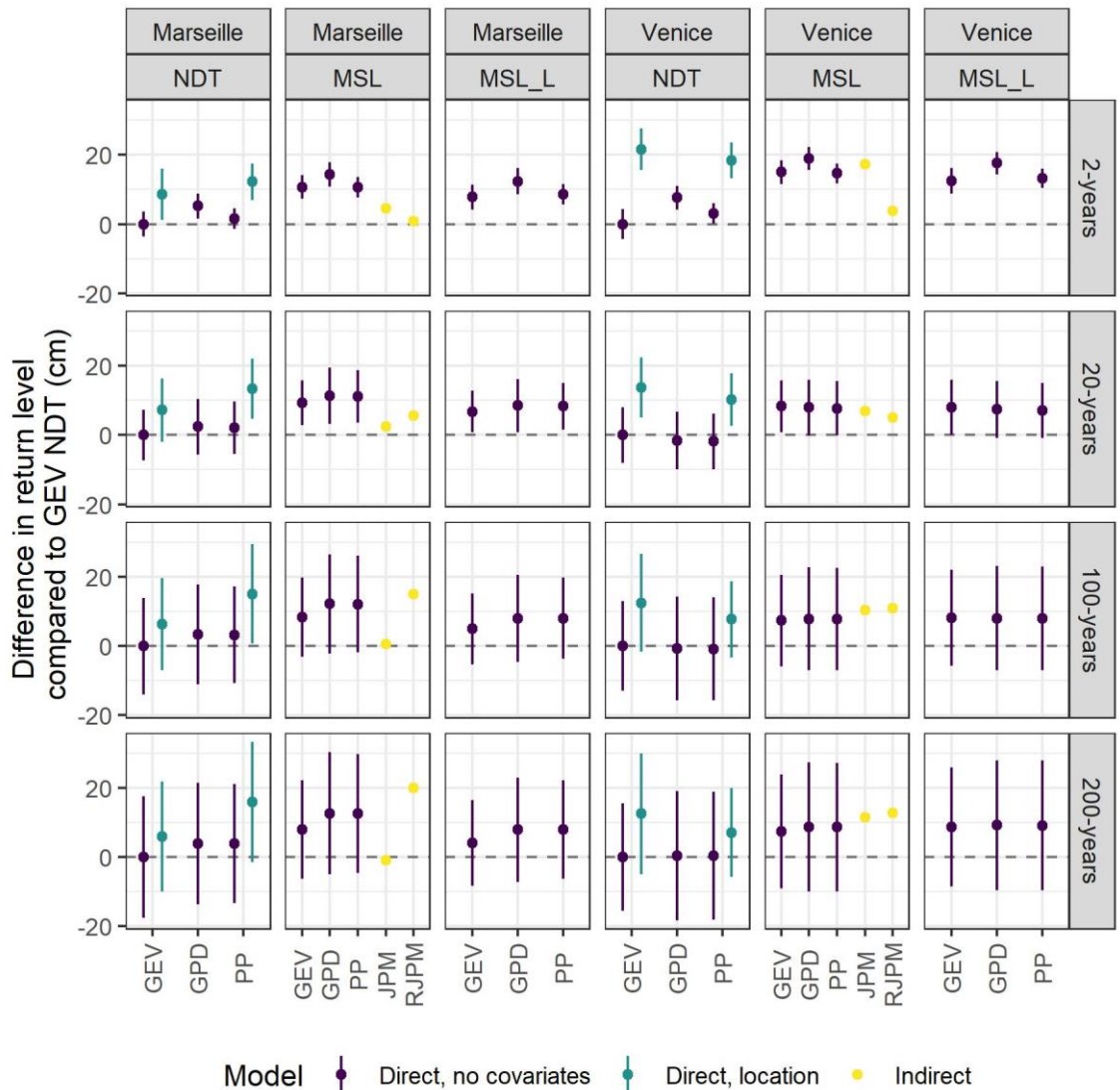

**570**

**Figure 5: Difference of return levels between each fitted model and a non-detrended, GEV fit for different return periods. Return levels of models with covariates are showed only if the model significantly improves the fit compared to models without covariates (p < 0.01; see Table 2). Plots are grouped vertically according to the detrending method (MSL: mean sea level, MSL_L: long term mean sea level, NDT: non detrended), and horizontally according to the distribution function (GEV: generalized extreme values,**
**575** **GP: generalized pareto, PP: point process).**

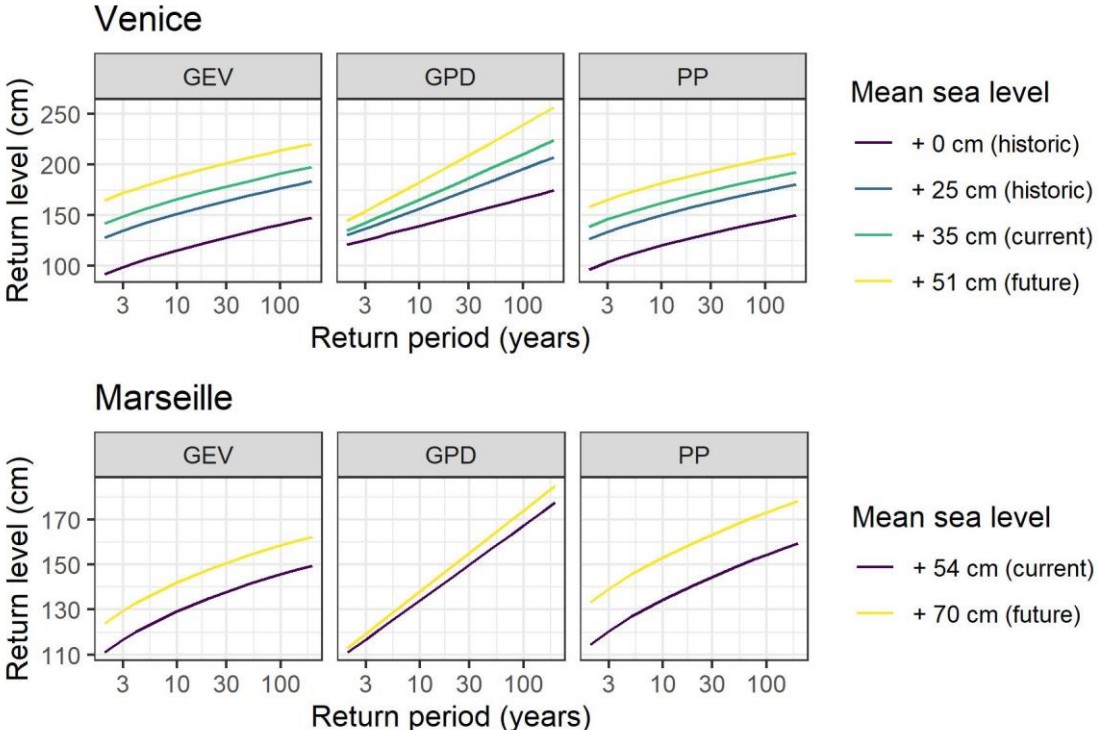

**Figure 6: Return level plots for different values of mean sea level. Mean sea level is expressed with respect to the local reference. Current mean sea level is + 35 for Venice and + 54 for Marseille. See section 2.4 for a description of the selected future mean sea levels.**

**Table 1: Trend in the data used to fit the models. lm: linear model; qr: quantile regression (0.75th data quantile).**

| Gauge | Extremes selection | Detrending | Regression Type | Test statistic | p-value | $R^2$ |
|---|---|---|---|---|---|---|
| VE | BM | NDT | lm | F(1,94) = 61.089 | p = 7.75 $10^{-12}$*** | 0.38 |
| VE | POT | NDT | lm | F(1,317) = 3.265 | p = 0.07. | 0.01 |
| VE | POT_q | NDT | qr | F(1,317) = 2.733 | p = 0.09. | - |
| MS | BM | NDT | lm | F(1,63) = 7.093 | p = 0.009** | 0.08 |
| MS | POT | NDT | lm | F(1,201) = 0.088 | p = 0.76n.s. | -0.004 |
| MS | POT_q | NDT | qr | F(1,201) = 0.217 | p = 0.64n.s. | - |
| VE | BM | MSL | lm | F(1,94) = 7.662 | p = 0.006** | 0.06 |
| VE | POT | MSL | lm | F(1,282) = 2.417 | p = 0.12n.s. | 0.005 |
| VE | POT_q | MSL | qr | F(1,282) = 1.102 | p = 0.29n.s. | - |
| MS | BM | MSL | lm | F(1,63) = 0.214 | p = 0.64n.s. | -0.01 |
| MS | POT | MSL | lm | F(1,206) = 0.001 | p = 0.98n.s. | -0.005 |
| MS | POT_q | MSL | qr | F(1,206) = 0.147 | p = 0.70n.s. | - |
| VE | BM | MSL_L | lm | F(1,94) = 14.276 | p = 0.00027*** | 0.12 |
| VE | POT | MSL_L | lm | F(1,357) = 5.432 | p = 0.020* | 0.01 |
| VE | POT_q | MSL_L | qr | F(1,357) = 5.058 | p = 0.025* | - |
| MS | BM | MSL_L | lm | F(1,63) = 0.207 | p = 0.65n.s. | -0.01 |
| MS | POT | MSL_L | lm | F(1,221) = 0 | p = 0.99n.s. | -0.004 |
| MS | POT_q | MSL_L | qr | F(1,221) = 0.463 | p = 0.49n.s. | - |

**Table 2: Likelihood ratio test results. The column test type describes which models configurations were compared: nc-l no covariates compared with covariates on location, l-sl: covariates on the location compared with covariates on both location and scale, nc-s no covariates compared with covariates on scale. VE = Venice; MS = Marseille.**

| Detrending | Distribution | Test Type | Test Statistic VE | p-value VE | Test Statistic MS | p-value MS |
|---|---|---|---|---|---|---|
| NDT | GEV | nc-l | $\chi_1 = 53.582$ | $2.48\ 10^{-13}$*** | $\chi_1 = 6.395$ | 0.011* |
| NDT | GEV | l-sl | $\chi_1 = 1.141$ | 0.28n.s. | $\chi_1 = 0.109$ | 0.74n.s. |
| NDT | GPD | nc-s | $\chi_1 = 3.958$ | 0.046* | $\chi_1 = 0.175$ | 0.67n.s. |
| NDT | PP | nc-l | $\chi_1 = 122.945$ | $1.43\ 10^{-28}$*** | $\chi_1 = 43.768$ | $3.69\ 10^{-11}$*** |
| NDT | PP | l-sl | $\chi_1 = 3.799$ | 0.051. | $\chi_1 = 0.346$ | 0.55n.s. |
| MSL | GEV | nc-l | $\chi_1 = 6.903$ | 0.008** | $\chi_1 = 0.379$ | 0.53n.s. |
| MSL | GEV | l-sl | $\chi_1 = 1.141$ | 0.28n.s. | $\chi_1 = 0.11$ | 0.73n.s. |
| MSL | GPD | nc-s | $\chi_1 = 3.358$ | 0.06. | $\chi_1 = 0.003$ | 0.95n.s. |
| MSL | PP | nc-l | $\chi_1 = 3.078$ | 0.079. | $\chi_1 = 0.123$ | 0.72n.s. |
| MSL | PP | l-sl | $\chi_1 = 3.086$ | 0.078. | $\chi_1 = 0.001$ | 0.97n.s. |
| MSL_L | GEV | nc-l | $\chi_1 = 13.887$ | $1.94\ 10^{-4}$*** | $\chi_1 = 0.099$ | 0.75n.s. |
| MSL_L | GEV | l-sl | $\chi_1 = 1.063$ | 0.30n.s. | $\chi_1 = 0.084$s. | 0.77n.s. |
| MSL_L | GPD | nc-s | $\chi_1 = 6.213$ | 0.012* | $\chi_1 = 0.005$ | 0.94n.s. |
| MSL_L | PP | nc-l | $\chi_1 = 13.42$ | $2.48\ 10^{-4}$*** | $\chi_1 = 7.087$ | 0.007** |
| MSL_L | PP | l-sl | $\chi_1 = 4.878$ | 0.027* | $\chi_1 = 0.601$. | 0.43n.s. |

**Table 3: Comparisons between the rates fitted by the point process (PP) for Venice and the empirical process rate of models with covariates on the location (Model Type = l), and models with covariates on location and scale (Model Type = s)**

| Detrending | Model Type | Test statistic | p-value | $R^2$ |
|---|---|---|---|---|
| NDT | l | $t(94) = 12.092$ | $p = 7.37 \ 10^{-21}$*** | 0.78 |
| NDT | ls | $t(94) = 12.344$ | $p = 2.22 \ 10^{-21}$*** | 0.78 |
| MSL | l | $t(94) = 1.8$ | $p = 0.07.$ | 0.18 |
| MSL | ls | $t(94) = 1.754$ | $p = 0.08.$ | 0.17 |
| MSL_L | l | $t(94) = 3.608$ | $p = 4.97 \ 10^{-4}$*** | 0.34 |
| MSL_L | ls | $t(94) = 3.451$ | $p = 8.39 \ 10^{-4}$*** | 0.33 |