# Peer review of "Importance of non-stationary analysis for assessing extreme sea levels under sea level rise"

_EGUsphere, 2022_

## Referee Comment (RC1)

Review of *Importance of non-stationary analysis for assessing extreme sea levels under sea level rise*

- Overview:

The manuscript compares the return levels obtained from four different extreme value analyses accounting for non-stationarity in the Punta Della Salute tide gauge, Venice. The extreme value distributions include: (i) generalized extreme value (GEV) applied to a block maxima sampling (BM); (ii) generalized Pareto distribution (GPD) and (iii) a point process (PP) method, both applied to peak over a threshold (POT); and (iv) a joint probability method (JPM). In addition, the authors tested the implications of using three different detrending techniques, including (i) removing the annual mean sea level from the time series (MSL); (ii) removing the last 19 years' mean sea level (MSL_L); and (iii) not detrending the data before fitting the distributions.

I find the topic important for coastal flood risk assessment as traditional designs have been based on analyses using direct methods that ignore the non-stationarities, which can lead to an underestimation of the risk, as found in previous studies. In addition, having a comprehensive analysis of the different non-stationary extreme value methodologies would help on the way to obtaining a more standardized analysis, facilitating the comparison of the results between the studies. Thus, the results of the manuscript are relevant for the scientific community and coastal risk stakeholders after improving the work in some aspects, mainly related to

- the level of comprehensiveness of the analysis: it will be interesting for the scientific community to include one of the most utilized indirect methods: the revised joint probability method. In doing so, the authors will also reduce the existing overlapping with previous studies,
- the level of applicability to other study areas by including more tide gauge records in the analysis, and
- the level of replicability (some information relevant for reproducibility is missed).

- General comments:

   1. The evaluation of return levels obtained from this set of methods in particular (GEV, GPD, PP, and JPM) has not been performed before, to the best of my knowledge. However, previous studies compared the implications

of using different extreme value distributions when accounting for non-stationarity (e.g., Razmi et al 2017; Menendez et al 2010). Thus, the paper will benefit from the analysis of a wider set of methodologies, particularly from including the revised joint probability method (RJPM), which is one of the most widely applied indirect methods for estimating non-stationary return levels.

In this line, some overlap also exists in the analysis of the effects of different detrending techniques in the estimation of return levels (Haigh et al 2010).

Finally, the uncertainties in estimating the parameters of the different extreme value distributions could be included in the analysis, as done in previous studies (e.g., Cheng et al 2014) to offer a more exhaustive analysis.

By doing all of the above, the paper will provide a richer and more novel analysis of the different non-stationary extreme value analyses, as well as a more complete evaluation of the uncertainties derived from each method.

2. As is, the manuscript conclusions are appropriate for the study case (Punta Della Salute) only. In order to provide a quasi-standardized method for non-stationary analysis, the paper will benefit from applying the analysis to a larger set of tide gauge records, so the authors will be able to assess whether the conclusions can be extrapolated to other areas of study. Likewise, readers will be able to decide which method should be used according to the conditions of each case: tide range, relative relevance of surge vs tide in extremes – as in Dixon et al 1999-, location, record length, etc. In this sense, Haigh et al 2010 showed that differences between return levels estimated using different methods highly depend on the record length and on the presence of outliers (when using direct methods).

3. In addition to the above comments, some concerns about the reproducibility of the work arise, including (i) the use of phrases such as "*We used our own script to generate the empirical sea level distribution function*" (line 151); (ii) the authors do not provide information on the additional tide gauges used to calculate mean sea level prior to 1924; (iii) they do not provide information on

how they calculated the uncertainties shown for the stationary analysis in 30-year time windows (Figure 3 and Figure 4).
* * *
- Specific questions

**Introduction**

- Line 58: How do you define suitability? What do you use to compare the different distributions and decide which of them performs best?

**Methods**

- Tide gauge: Is the sea level record at this particular tide gauge affected by the activation of the MOSE? Can that influence the measured extreme sea levels?

- Tide gauges used to calculate the mean sea level prior to 1924: (i) in order to ensure reproducibility, the authors should provide the name and location of those tide gauges. (ii) Are these extra tide gauges affected by the same subsidence rate as Punta Della Salute? As a double-check, did you compare the mean sea level of the tide gauges and the Punta Della Salute one during the overlapping period?

- Joint probability method: instead of using JPM, the authors might consider the revised JPM (RJPM) or the skew surge revised JPM (SSRJPM). These two methods allow the fitting of extreme value distribution (thus obtaining extreme probabilities) and apply an extreme index that accounts for dependence between extreme events.

Line 151: to ensure reproducibility you might want to clarify which kind of empirical distribution you have used instead of stating sentences such as "*we use our own script to generate the empirical sea level distribution*".

- Comparing different model configurations: Could you please expand the explanation of the meaning of $M_0$ and $M_1$?

**Results**

- Figure 3: I don't think you explained how you calculated the gray envelope in the text. The analysis will benefit from the comparison of uncertainties resulting from the different model configurations. Also, why isn't the scale parameter (yellow line) shown for GEV,

GPD (MSL), and PP? doesn't the scale parameter vary with time (equation 7)? Same for the location parameter (green line) for PP (NDT, MSL, MSL_L)?

- References

Haigh I D, Nicholls R, Wells N (2010). *A comparison of the main methods for estimating probabilities of extreme still water levels.* Coastal Engineering, vol 57 (9).

Dixon M J, Tawn J A (1999). *The effect of non-stationarity on extreme sea-level estimation.* Appl Statist (48).

Cheng L, AghaKouchak A, Gilleland E, Katz R W (2014). *Non-stationary extreme value analysis in a changing climate.* Climate Change.

Menendez M, Woodworth P L (2010). *Changes in extreme high water levels based on a quasi-global tide-gauge data set.* Journal of Geophysical Research – Oceans.

Razmi A, Golian A, Zahmatkesh Z (2017). *Non-Stationary Frequency Analysis of Extreme Water Level: Application of Annual Maximum Series and Peak-over Threshold Approaches.* Water Resources Management (31).

---

## Referee Report (RR1)

**Second review of "Importance of non-stationary analysis for assessing extreme sea levels under sea level rise"**

The authors have replied to my previous comments and have carried out the appropriate changes. In this second review, I'm providing minor comments related to the lack of explanation in some parts of the paper (mainly methods) and grammar. However, I may note two major comments (1 and 2) to the authors and the editor:

(1) The authors claim that one of the objectives of the present work is to assess "*which parametric method best accommodates non-stationarity conditions*". The discussion and conclusion sections are written as if the results can be extrapolated to any other location (except for one sentence in line 338). However, the non-stationary conditions change with the conditions of the area of study. As I noted in the first review of this paper: "*As is, the manuscript conclusions are appropriate for the study case (Punta Della Salute) only. In order to provide a quasi-standardized method for non-stationary analysis, the paper will benefit from applying the analysis to a larger set of tide gauge records, so the authors will be able to assess whether the conclusions can be extrapolated to other areas of study. Likewise, readers will be able to decide which method should be used according to the conditions of each case: tide range, relative relevance of surge vs tide in extremes – as in Dixon et al 1999-, location, record length, etc. In this sense, Haigh et al 2010 showed that differences between return levels estimated using different methods highly depend on the record length and on the presence of outliers (when using direct methods).*"

In the second version, the authors included a new location, the Marseille tide gauge, which indeed shows that results change from one location to another (lines 246 to 247; 271 to 272). The inclusion of this second tide gauge is not, however, sufficient to generalize the results. The comparison of different extreme value distributions has been performed previously. The novelty of this work relies on the fact that they use a set of methods that have not been compared yet (to the best of my knowledge). Although this provides some novelty, I strongly believe that the paper can make a difference in the literature by performing the analysis using a comprehensive set of locations (globally ideally).

Concluding, I suggest the authors choose one of two options 1) a local study in which the results are limited to the two tide gauges used. If this option is chosen, the narrative of the paper should be modified to reflect this, or; 2) include a full set of tide gauges so that differences in conditions (record length, tidal range, surge vs tide, etc.) are included in the analysis and the results can be extrapolated.

(2) The main objective of the paper is to evaluate the performance of various extreme value distributions to account for non-stationarity, so it is very methodologically focused. Acknowledging this, the exposition of the methods is short and poor. A discussion of the

pros and cons of each method would fit the paper. Also, the uncertainties of the models should be discussed in the results and conclusions.

(3) The article will benefit from a rewrite to avoid the "telegram style" taking particular attention to punctuation and typos, which are frequent throughout the paper (e.g., lines 247, 278, 299). It will also benefit from the integration into the narrative of the new tide gauge results. As is, the Marseilles tide gauge seems to be shoehorned into the paper.

**Abstract**

- Lines 6 and 7 need to be referenced. Coastal flooding is a very important hazard derived from mean sea level rise, however, many other climate change-related hazards pose important impacts (e.g., temperature). Also, the storm surge intensification has caused some debate in the literature.
- The abstract needs to be updated to include the results from the Marseille tide gauge. For instance, line 9 states that 96 years of data is used, which is not true for the case of the Marseille tide gauge (line 103).
- Line 15. What do you mean by "Actualized"?

**Introduction**

- Lines 34 to 41. The authors start the saying: "*Assuming stationarity when data are non-stationarity has several practical implications. First…*" However, they only indicate one: the use of return levels for designing structures.
  Also, the information contained in lines 37 to 42 is pretty much the same. I recommend to re-write this part to avoid redundancies and empty wording (i.e., a message that seems to contain meaningful content but does not).
- Lines 42 to 50. This is a miscellaneous paragraph:
  (i)    It starts with "*Several methods were proposed to cope with non-stationary conditions*" but the authors do not offer an introduction to these methods.
  (ii)   It continues with the importance of long-term records to identify non-stationarity. However, they don't include further information on that, for instance, what is the minimum record length to assess non-stationary conditions.
  (iii)  It highlights that mean sea level is not the only source of non-stationarity. Aren't the authors considering mean sea level as the only source of non-stationarity as well? Also, the different sources of non-stationarity are indicated here and on lines 33 to 34.
  (iv)   It is stated that there are no "*clear indications on which approach suits better non-stationary conditions*". What do you mean by "approaches"; the detrending techniques, the extreme value distributions (which you speak about in the next

paragraph as a second challenge), the extreme value parameters that vary with mean sea level?

I believe this paragraph and also the introduction section will benefit from an in-depth revision from the authors.

- Line 61: "*Given the above knowledge gaps*": Depending on the definition of "approach" (see my comment above), I only see one gap: identify the best model to account for non-stationarity (derived from mean sea level) in extreme sea levels from the set of models presented in the paper.

**Methods**

- Lines 95 to 96. Do you interpolate the half-hour and 10 minutes data to hourly data? Working with different time resolution data can influence your results.
- Line 125. How do you define "stability" in this case? More explanation should be provided to ensure the reproducibility of your analysis.
- Line 145: "*This property can drive the selection of an appropriate threshold u*", how?
- Lines 153 to 154. I would extend this explanation a bit more. What message do you want to deliver to the reader by saying that the PP is called nonhomogeneous when you include covariates in your model?
- Line 161. You might want to use a better reference for the independency between surges and tides. Marcos et al 2009 just describe the method they use in their paper but do not provide a theoretical background for that assumption.
- Lines 163 to 164. Other works have used the RJPM fitting the extreme value distribution to the values above a threshold (Batstone et al., 2013; Baranes et al., 2020; Enríquez et al., 2022) instead of using the highest measured surge.
- Line 170. Wöppelmann et al (2014) found 24 main tidal constituents in Marseille (Table 2 in their paper), since you are citing them here, why do you use 21 tidal constituents, and what are those?
- Line 174. The calculation of the empirical return period in the RJPM is still unclear.
- The RJPM needs further explanation.

**Results**

- Line 249 to 250. "*The location is included in the scale parameter of the GOD that does not improve the fit (Fig. 3)*". This needs more explanation.
- Lines 254 to 259. Are these results for Punta della Salute only or for both locations?
- The uncertainties are not discussed in the Results.
- A similar figure as Fig. 3 should be included for Marseille.
- Yellow area in the figures are difficult to see.
- Some reasons why the analyses of Marseille tide gauge seem to be shoehorned in the paper:

1.  Marseille results are missed in the Abstract section
2.  Lines 63 to 65
3.  Section 2.1: 17 lines speaking about the Venice lagoon and the Punta della Salute station versus almost 2 lines about Marseille.

**Conclusions**

Conclusion section is too short. Which extreme value model should we use at the two tide gauges if we want to analyze non-stationarity? Which one result in less uncertainty?

---

## Referee Report (RR2)

**Third review of "Importance of non-stationary analysis for assessing extreme sea levels under sea level rise"**

Around line 45. Nice explanation overall. I do think there is an overuse of brackets that hamper the reading. I would separate out the sampling methods (peaks over threshold and block maxima) from the extreme value distributions (GEV, PP, GPD).

Line 57. Isn't the comparison of different detrending techniques another objective?

Around lines 80. *We resampled all data recorded after 1989 to an hourly resolution with different Pugh filters*. Did you use more than one filter to resample the data? If not, which one did you use? Please, think about reproducibility.

Line 90. Typo

Line 157. where $= x + y$. Is this a typo?

Lines 173-174. *We used 1990-2019 hourly data from for Venice and 1968-2016 for Marseille (record length of 30 years for both stations).* Are these the years you have used to calculate the tidal coefficients? Please, indicate.

Line 183. What do these numbers mean?

Line 226. I don't understand Cheng et al 2014 reference here. You are talking about what you did.

Line 256. *When, where*

Results section is surprisingly short.

Line 304 to 308. Is this information based on your analysis? What results are you using to claim these conclusions?

Line 368- 369. *Overall, we show that using different methods allows to critically examine strengths and weaknesses of each method and to critically evaluate the results to drive the choice of the method that best fits the specific case.* This is the objective of the present paper, right? I would remove the sentence from your paper. What valuable information are you providing by saying this?

---

## Author Response (AR2)

Dear Editor,

We are thankful to the anonymous reviewer for the constructive comments and the appreciation of our work.

Based on the comments from the reviewer, we revised the paper in the following aspects. (1) We adjusted aims and objectives of the paper, stressing that this paper aims at providing the reader a list of the most used methods for extreme sea levels assessments, and the way they can cope with non stationary conditions. We also stressed that as a proof of applicability we implement and compare the methods to data from gauging stations with different extents of non-stationarity. (2) given the new aims of the paper, we extended the description of the methods and we improved it following the reviewer's suggestions. (3) We extended the results, the discussion, and the conclusions to better include the Marseille dataset in the paper's narrative. We also extended the discussion to better show strengths and limitations of each method.

All the points raised by the reviewer were addressed. Detailed, point-to-point changes are listed in the attached file.

We hope that these changes will increase the impact of the study and are happy to hear back from you in due time.

Sincerely,
Damiano Baldan,
on behalf of all co-authors

**General comments**
**Answers marked in blue**

The authors have replied to my previous comments and have carried out the appropriate changes. In this second review, I'm providing minor comments related to the lack of explanation in some parts of the paper (mainly methods) and grammar. However, I may note two major comments (1 and 2) to the authors and the editor:

(1) The authors claim that one of the objectives of the present work is to assess "*which parametric method best accommodates non-stationarity conditions*". The discussion and conclusion sections are written as if the results can be extrapolated to any other location (except for one sentence in line 338). However, the non-stationary conditions change with the conditions of the area of study. As I noted in the first review of this paper: "*As is, the manuscript conclusions are appropriate for the study case (Punta Della Salute) only. In order to provide a quasi-standardized method for non-stationary analysis, the paper will benefit from applying the analysis to a larger set of tide gauge records, so the authors will be able to assess whether the conclusions can be extrapolated to other areas of study. Likewise, readers will be able to decide which method should be used according to the conditions of each case: tide range, relative relevance of surge vs tide in extremes – as in Dixon et al 1999-, location, record length, etc. In this sense, Haigh et al 2010 showed that differences between return levels estimated using different methods highly depend on the record length and on the presence of outliers (when using direct methods).*"

In the second version, the authors included a new location, the Marseille tide gauge, which indeed shows that results change from one location to another (lines 246 to 247; 271 to 272). The inclusion of this second tide gauge is not, however, sufficient to generalize the results. The comparison of different extreme value distributions has been performed previously. The novelty of this work relies on the fact that they use a set of methods that have not been compared yet (to the best of my knowledge). Although this provides some novelty, I strongly believe that the paper can make a difference in the literature by performing the analysis using a comprehensive set of locations (globally ideally).

Concluding, I suggest the authors choose one of two options 1) a local study in which the results are limited to the two tide gauges used. If this option is chosen, the narrative of the paper should be modified to reflect this, or; 2) include a full set of tide gauges so that differences in conditions (record length, tidal range, surge vs tide, etc.) are included in the analysis and the results can be extrapolated.

Thank you for the comment. We acknowledge that the comparison of methods carried out in the paper cannot be generalized. We acknowledge the soundness of the proposal from the reviewer to repeat our analysis for a global dataset. We also believe such in-depth analyses would deserve a dedicated publication, as it would require a significant restructuring of both the paper structuring and the analyses carried out so far, including the retrieval of a validated global tide sea level dataset. Thus, we decided to follow the option 1) indicated by the reviewer, i.e. to modify the narrative of the paper.

(2) The main objective of the paper is to evaluate the performance of various extreme value distributions to account for non-stationarity, so it is very methodologically focused. Acknowledging this, the exposition of the methods is short and poor. A discussion of the pros and cons of each method would fit the paper. Also, the uncertainties of the models should be discussed in the results and conclusions.

Thank you for the comment. We acknowledge that some explanations of the methods were too short, and we extended the methods section following the detailed comments from the reviewer to improve the

reproducibility. We also included some sentences in the discussion with the strengths and limitations of each method.

(3) The article will benefit from a rewrite to avoid the "telegram style" taking particular attention to punctuation and typos, which are frequent throughout the paper (e.g., lines 247, 278, 299). It will also benefit from the integration into the narrative of the new tide gauge results. As is, the Marseilles tide gauge seems to be shoehorned into the paper.

Thank you for the comment. We extended the methods, results, discussion, and conclusion sections to better include the Marseille dataset in the narrative. We also carefully checked the manuscript for inappropriate punctuation and typos.

**Specific comments**

| Rewiewer's comment | Action taken | Lines |
|---|---|---|
| **Abstract** | | |
| Lines 6 and 7 need to be referenced. Coastal flooding is a very important hazard derived from mean sea level rise, however, many other climate change-related hazards pose important impacts (e.g., temperature). Also, the storm surge intensification has caused some debate in the literature. | Thank you for the comment. To remove ambiguity in the text we shortened the sentence. Now it reads:

"Increased coastal flooding caused by extreme sea levels (ESLs) is one of the major hazards related to sea level rise."

The concept is then referenced in the introduction. | 6-7 |
| The abstract needs to be updated to include the results from the Marseille tide gauge. For instance, line 9 states that 96 years of data is used, which is not true for the case of the Marseille tide gauge (line 103). | Thanks for the comment. We included the reference to Marseille in the abstract:

"In this work, we fit several extreme values models to two long-term sea level record from Venice (96 years) and Marseille (65 years)" | 9 |
| Line 15. What do you mean by "Actualized"? | Thanks for the comment. This word is explained in the main text, but we feel there is not enough space in the abstract for explaining, so we removed it. | - |
| **Introduction** | | |
| Lines 34 to 41. The authors start the saying: "*Assuming stationarity when data are non-stationarity has several practical implications. First…*" However, they only indicate one: the use of return levels for designing structures. | Thanks for the comment. We rewrote the paragraph to account for the observation, see answers to the following comments. | |
| Also, the information contained in lines 37 to 42 is pretty much the same. I recommend to re-write this part to avoid redundancies and empty wording (i.e., a message that seems to contain meaningful content but does not). | Thanks for the comment. We rewrote and shortened the paragraph to remove redundant information:

"The results of extreme value theory are valid under the assumptions of independence and stationarity of extremes (Khaliq et al., 2006). Here, stationarity means that all the realizations of the extremes in the data record are generated from the same distribution (Coles et al., 2001). While independence is satisfied with a proper selection of extremes from the dataset, stationarity is often assumed but not verified (Khaliq et al., 2006). However, several sources of non-stationarity can affect sea | 28 - 36 |

| | level data: changes in coastal morphology, low frequency climatic variability, and climate change (Salas and Obeysekera, 2014). The estimation of return levels from stationary models might not be appropriate (e.g. less conservative) because of the implicit assumptions that the characteristics of the extremes remains the same in the future (Caruso and Marani, 2022; Razmi et al., 2017; Dixon and Tawn, 1999; Salas and Obeysekera, 2014; Haigh et al., 2010; Ragno et al., 2019)." | |
|---|---|---|
| Lines 42 to 50. This is a miscellaneous paragraph: (i) It starts with "*Several methods were proposed to cope with non-stationary conditions*" but the authors do not offer an introduction to these methods. (ii) It continues with the importance of long-term records to identify non-stationarity. However, they don't include further information on that, for instance, what is the minimum record length to assess non-stationary conditions. (iii) It highlights that mean sea level is not the only source of non-stationarity. Aren't the authors considering mean sea level as the only source of non-stationarity as well? Also, the different sources of non-stationarity are indicated here and on lines 33 to 34. (iv) It is stated that there are no "*clear indications on which approach suits better non-stationary conditions*". What do you mean by "approaches"; the detrending techniques, the extreme value distributions (which you speak about in the next paragraph as a second challenge), the extreme value parameters that vary with mean sea level?

I believe this paragraph and also the introduction section will benefit from an in-depth revision from the authors. | Thanks for the comment. Since the focus of this paragraph is on methods to account for non-stationarity, we rewrote the paragraph removing the redundant information, and joined with the previous one:

"Two approaches are commonly used to cope with non-stationarity. Detrending the sea level data with annual or long term mean sea levels is a common practice to remove long term signals in the mean of the dataset (Bernier et al., 2007; Tebaldi et al., 2012; Mentaschi et al., 2016). Alternatively, the parameters of the probability distribution function that generates the extremes can be explicitly modelled as dependent from some covariates (Méndez et al., 2007; Grinsted et al., 2013; Cid et al., 2016; Sweet and Park, 2014; Razmi et al., 2017). However, clear indications on which approach suits better non-stationary conditions are still missing." | 36 - 41 |
| - Line 61: "*Given the above knowledge gaps*": Depending on the definition of "approach" (see my comment above), I only see one gap: identify the best model to account for non-stationarity (derived from mean sea level) in extreme sea levels from the set of models presented in the paper. | Thank you for the comment. We rewrote the objectives and aims paragraph to adjust the narrative of the paper:

"Using two long-term sea level time series from Venice (96 years, NE Italy), and Marseille (65 years, Southern France) with different extents of non-stationarity, this this | 53 - 57 |

| | paper aims at: (i) compiling information on the existing direct and indirect methods for extreme sea levels estimation; (ii) assessing which parametric method best accommodates non-stationary conditions; and (iii) comparing return level and return period estimates from different parametric and non-parametric methods. We perform all the analyses using three different detrending approaches." | |
|---|---|---|
| **Methods** | | |
| Lines 95 to 96. Do you interpolate the half-hour and 10 minutes data to hourly data? Working with different time resolution data can influence your results. | Thanks for the comment. We used Pugh filters to resample the data to a hourly time resolution. We added this indication in the text:

"We resampled all data recorded after 1989 to an hourly resolution with different Pugh filters (Pugh, 1987)." | 82 - 83 |
| Line 125. How do you define "stability" in this case? More explanation should be provided to ensure the reproducibility of your analysis. | Thanks for the comment. We added the explanation in section 2.3.2 (we feel it fits better there), and pointed to that section here –see answer to the next comment. | 114 |
| - Line 145: "*This property can drive the selection of an appropriate threshold u*", how? | Thanks for the comment. We added an explanation and included a reference.

"This property can drive the selection of an appropriate threshold u. First, multiple GPD distributions are fitted to different sets of data obtained varying the threshold. Then, the parameters are plotted as a function of the threshold. For sufficiently high thresholds, the theoretical approximation yields and the parameters are independent of the threshold value. The minimum threshold that meets this requirement is then selected (Coles et al., 2001)" | 137 - 140 |
| Lines 153 to 154. I would extend this explanation a bit more. What message do you want to deliver to the reader by saying that the PP is called nonhomogeneous when you include covariates in your model? | Thanks for the comment. What we want to explain is that the PP is particularly suitable for extremes whose occurrence frequency changes with some covariates. We added this explanation and a reference:

"When location and scale are not constant (e.g. a dependence from a covariate is introduced), the process rate is not constant over time and the point process is non | 148 - 151 |

| | homogeneous (Cebrián et al., 2015). Accordingly, the probability of occurrence of extremes in a non-homogeneous point process is not constant over time, hence this model is appropriate also for modelling extremes whose occurrence frequency is not constant in time (Coles et al., 2001)." | |
|---|---|---|
| Line 161. You might want to use a better reference for the independency between surges and tides. Marcos et al 2009 just describe the method they use in their paper but do not provide a theoretical background for that assumption. | Thanks for the comment. We modified the sentence and added a more appropriate reference on tide-surge interaction.

"The tide and the surge interaction is significant only in shallow waters (Prandle and Wolf, 1978). Therefore, in most applications, tide and surge can be considered independent." | 158 - 160 |
| Lines 163 to 164. Other works have used the RJPM fitting the extreme value distribution to the values above a threshold (Batstone et al., 2013; Baranes et al., 2020; Enríquez et al., 2022) instead of using the highest measured surge. | Thanks for the comment. We included in the text that also POT approaches can be used to fit the surge in the RJPM, and included the mentioned references:

"Second, the RJPM fits the surge distribution with an extreme values probability distribution function to smooth the empirical distribution, and for projections beyond the highest measured surge (Tawn et al., 1989). Both GEV and GPD have been used to this end (Baranes et al., 2020; Batstone et al., 2013; Enríquez et al., 2022)." | 166 - 168 |
| Line 170. Wöppelmann et al (2014) found 24 main tidal constituents in Marseille (Table 2 in their paper), since you are citing them here, why do you use 21 tidal constituents, and what are those? | Thanks for the comment. We corrected the typo and reported the names of the tide constituents:

"[…] and 24 harmonic constants for Marseille (MSM, MM, MSF, MF, Q1, O1, NO1, PI1, P1, S1, K1, J1, 2N2, MU2, N2, NU2, M2, L2, T2, S2, K2, MN4, M4, MS4; Wöppelmann et al., 2014)." | 170 - 172 |
| Line 174. The calculation of the empirical return period in the RJPM is still unclear. | Thanks for the comment. We extended the explanation in section 2.3.7 to include also JPM and RJPM in the calculation of return periods:

"The return period is defined as: $Tr(z)=[1-G(z)]^{-1}$, where G is the Probability Distribution Function for the GEV, GPD, or PP models (Caruso and Marani, 2022), or the | |

| | empirical sea level probability from the JPM and RJPM (Tawn et al., 1989)." | |
|---|---|---|
| The RJPM needs further explanation. | We extended the RJPM explanation by stressing the way it improves the JPM:

"Two limitations of the JPM are that consecutive sea levels are assumed to be independent, and that the upper tail of the empirical surge distribution is biased by the lack of observations of extremes. As a result, the JPM cannot produce ESLs estimates for sea levels higher than the combination of the highest tide and surge (Batstone et al., 2013; Tawn et al., 1989). The revised Joint Probability Method (RJPM) aims at improving both issues. First, an extremal index that accounts for dependencies in the sea level data is introduced. The extremal index is used as a correction factor in the return period calculation based on P(z) (see section 2.3.7), and is defined as the average number of measurements an extreme sea level cluster is usually composed of (Tawn et al., 1989). Second, the RJPM fits the surge distribution with a extreme values, to allow for the smoothing of the empirical distributions, and for projections beyond the highest measured surge (Tawn et al., 1989). Both GEV and GPD have been used to this end (Baranes et al., 2020; Batstone et al., 2013; Enríquez et al., 2022)." | 207 - 208 |
| **Results** | | |
| Line 249 to 250. "*The location is included in the scale parameter of the GOD that does not improve the fit (Fig. 3)*". This needs more explanation. | Thanks for the comment. We think this sentence pertains more to the discussion section and therefore we deleted it from the results section. | - |
| Lines 254 to 259. Are these results for Punta della Salute only or for both locations? | Thanks for the comment. We expanded and rewrote the paragraph to present results from both gauging stations more clearly. | 265 - 268 |
| The uncertainties are not discussed in the Results. | Thanks for the comment. We included a paragraph in the results that discusses the uncertainties in the direct methods:

"The direct methods show varying uncertainty in the prediction of return levels (Figure 5). In both Venice and Marseille, the | 283 - 289 |

| | GEV with covariates on the location has the highest uncertainty (12 cm in Venice and 15 cm in Marseille) for the 2 years return period, and the PP without covariates the lowest uncertainty (7 cm both in Venice and Marseille). In Venice, the PP with covariates on the location has the lower uncertainty for return levels of 20, 100, and 200 years (15, 20, and 25 cm, respectively). Non detrended models without covariates and detrended models have similar uncertainty (slightly lower for GEV). In Marseille, the GEV fitted to non detrended data has the lowest uncertainty for return levels of 20, 100, and 200 years (13, 23, and 27 cm, respectively)." | |
|---|---|---|
| A similar figure as Fig. 3 should be included for Marseille. | Thanks for the comment. We generated a similar figure for Marseille data. We decided to include this figure in the supplementary material to avoid an excessive number of figures in the main text. | fig 3 |
| Yellow area in the figures are difficult to see. | We changed the transparency of the colors in the figures to improve readability. | fig 3 |
| Some reasons why the analyses of Marseille tide gauge seem to be shoehorned in the paper:
1. Marseille results are missed in the Abstract section
2. Lines 63 to 65
3. Section 2.1: 17 lines speaking about the Venice lagoon and the Punta della Salute station versus almost 2 lines about Marseille. | We added some more text on Marseille in the text and in the abstract.

"In this work, we fit several extreme values models to two long-term sea level record from Venice (96 years) and Marseille (65 years) […]"

"On the contrary, the area where the Marseille tide gauge is located has a lower tidal range (around 10 cm, Fig S1), and is located on a stable geological background, with a relative sea level rise of + 1.1 mm $y^{-1}$ in the last 150 years (Letetrel et al., 2010; Wöppelmann et al., 2014). Marseille data are referred to the nautical chart datum (NCD, Zero Hydrographique), the sea level corresponding to the lowest tide, and is 32.9 cm below the national datum (IGN1969, average sea level for the period 1885-1897 measured at Marseille gauging station, Wöppelmann et al., 2014). The long term mean sea level was 35 cm above the NCD in 1903 and 50 cm above the NCD in 2017." | - |

| Conclusions | | |
|---|---|---|
| Conclusion section is too short. Which extreme value model should we use at the two tide gauges if we want to analyze non-stationarity? Which one result in less uncertainty? | Thanks for the comment. We included the information in the conclusion.

"In this paper, we fitted different extreme value models to long-term sea level data for Venice and Marseille. We show that including non-stationarity in the analysis of extreme events improves the fit of most of the models. Among direct methods, for return periods longer than 20 years, the Point Process with a dependence of the location from the mean sea level is the most conservative in Venice. The Generalized Extreme Values distribution with a dependence of the location from the mean sea level is the most conservative in Marseille. Among indirect methods, the Revised Joint Probability Method yields results that are comparable with the most conservative methods for return periods longer than 100 years for both Venice and Marseille. Among direct methods, the Generalized Extreme Values Distribution fitted to detrended data has the lowest uncertainty for return levels estimation in Venice. The Point Process with a location dependence has the lowest uncertainty for return levels estimation in Marseille for return periods longer than 20 years. Overall, we show that non-stationary extremes analyses can provide more robust estimates of return levels to be used in coastal protection planning." | 372 - 379 |